# Impact of Guideline Adherence on Outcomes in Patients Hospitalized with Community-Acquired Pneumonia (CAP) in Hungary: A Retrospective Observational Study

**DOI:** 10.3390/antibiotics11040468

**Published:** 2022-03-30

**Authors:** Adina Fésüs, Ria Benkő, Mária Matuz, Zsófia Engi, Roxána Ruzsa, Helga Hambalek, Árpád Illés, Gábor Kardos

**Affiliations:** 1Central Clinical Pharmacy, Clinical Center, University of Debrecen, H-4032 Debrecen, Hungary; fesus.adina@pharm.unideb.hu; 2Department of Pharmacodynamics, Faculty of Pharmacy, University of Debrecen, H-4032 Debrecen, Hungary; 3Doctoral School of Pharmaceutical Sciences, University of Debrecen, H-4032 Debrecen, Hungary; 4Health Industry Competence Centre, University of Debrecen, H-4032 Debrecen, Hungary; 5Clinical Pharmacy Department, Faculty of Pharmacy, University of Szeged, H-6725 Szeged, Hungary; benko.ria@med.u-szeged.hu (R.B.); matuz.maria@szte.hu (M.M.); engi.zsofia@szte.hu (Z.E.); roxana.ruzsa@gmail.com (R.R.); helgahambalek@gmail.com (H.H.); 6Central Pharmacy, Albert Szent Györgyi Medical Center, University of Szeged, H-6725 Szeged, Hungary; 7Department of Emergency Medicine, Albert Szent Györgyi Medical Center, University of Szeged, H-6725 Szeged, Hungary; 8Department of Internal Medicine, Faculty of Medicine, University of Debrecen, H-4032 Debrecen, Hungary; illes.arpad@med.unideb.hu; 9Department of Metagenomics, University of Debrecen, H-4032 Debrecen, Hungary

**Keywords:** community acquired pneumonia, hospitalized patients, empirical antibiotic therapy, guideline adherence, clinical outcomes, 30-day mortality, CRP on admission, CCI score

## Abstract

Community-acquired pneumonia (CAP) is a leading cause of morbidity and mortality worldwide. This retrospective observational study evaluated the antibiotic prescription patterns and associations between guideline adherence and outcomes in patients hospitalized with CAP in Hungary. Main outcome measures were adherence to national and international CAP guidelines (agent choice, dose) when using empirical antibiotics, antibiotic exposure, and clinical outcomes. Demographic and clinical characteristics of patients with CAP in the 30-day mortality and 30-day survival groups were compared. Fisher’s exact test and t-test were applied to compare categorical and continuous variables, respectively. Adherence to the national CAP guideline for initial empirical therapies was 30.61% (45/147) for agent choice and 88.89% (40/45) for dose. Average duration of antibiotic therapy for CAP was 7.13 ± 4.37 (mean ± SD) days, while average antibiotic consumption was 11.41 ± 8.59 DDD/patient (range 1–44.5). Adherence to national guideline led to a slightly lower 30-day mortality rate than guideline non-adherence (15.56% vs. 16.67%, *p* > 0.05). In patients aged ≥ 85 years, 30-day mortality was 3 times higher than in those aged 65–84 years (30.43% vs. 11.11%). A significant difference was found between 30-day non-survivors and 30-day survivors regarding the average CRP values on admission (177.28 ± 118.94 vs. 112.88 ± 93.47 mg/L, respectively, *p* = 0.006) and CCI score (5.71 ± 1.85 and 4.67 ± 1.83, *p* = 0.012). We found poor adherence to the national and international CAP guidelines in terms of agent choice. In addition, high CRP values on admission were markedly associated with higher mortality in CAP.

## 1. Introduction

The use of antibiotics has significantly reduced bacterial infection-related morbidity and mortality; their inappropriate use, however, has led to the emergence of antibiotic resistance at the same time [1,2].

The epicenter of antibiotic resistance is the hospital environment; thus, it is critical to rationalize antibiotic use in this setting. In European acute-care hospitals, 35% of patients receive systemic antibiotics during their stay [3]. European Centre for Disease Prevention and Control (ECDC) point prevalence survey data showed that antibiotic use for community acquired infections (CAIs) in Europe represented 69.9% of all antibacterial use in acute-care hospitals [4]. In particular, more than one-third (35%) of CAIs were found to be respiratory tract infections (RTIs) [5]. In 2016, respiratory illnesses were the third most common cause of death in Europe, and accounted for 7.5% of all deaths, while in 2017 the corresponding number in Hungary was 6.2% [6]. Community-acquired pneumonia (CAP) is one of the most common and potentially serious infectious diseases and is still one of the leading causes of morbidity and mortality worldwide [7,8,9], imposing a heavy economic burden on health systems even in developed countries. In European countries, CAP was responsible for almost 30% of mortality in the category of respiratory illnesses in 2015 [10]. Although CAP is often treated in ambulatory settings, hospitalization rates range from 30% to 60% [11]. Recent CDC data found that in the United States, 79% of all patients with CAP were treated inappropriately in the hospital setting [12]. Inappropriateness of hospital treatment of CAP is associated with worse therapy outcomes, longer hospital stays, and higher cost of treatment [13,14,15,16,17]. The National Institute for Health and Care Excellence and British Thoracic Society (NICE/BTS) and the American Thoracic Society and the Infectious Diseases Society of America (ATS/IDSA) have published official clinical CAP guidelines making recommendations for selection of initial empiric antibiotic therapy for patients hospitalized with CAP. A national CAP guideline has also been published by the Hungarian Professional Society of Infectious Diseases and Pulmonology.

To date, descriptions of antibiotic treatment trends for CAP have only been published for adult outpatient care in Hungary [18]. Despite the importance and incidence of CAP, no field studies have been performed in Hungarian hospitals to assess the initiated antibiotic treatments.

The aim of this study was to evaluate the characteristics and outcome of antibacterial drug use in patients admitted to hospital due to CAP. The primary aims were to evaluate adherence to national and international antibacterial guidelines and to analyze the potential factors associated with mortality. Secondly, we reported some basic characteristics of antibacterial treatments used in CAP.

## 2. Results

In the study period, data of 1665 patients were collected, out of which data obtained from 147 patients met the study criteria and could be included in the analysis.

### 2.1. Patient Characteristics and Main Outcomes

The characteristics of patients and their comorbidities are described in Table 1. A total of 64 (43.54%) male patients hospitalized due to CAP were included in the study. Their age at hospital admission ranged from 27 to 95 years; 118 (80.27%) patients were aged ≥ 65 years (Table 1). Overall, 59.86% of patients had a CCI score above 4. The most common comorbidities included cardiovascular diseases (35.37%) and diabetes mellitus (22.45%) (Table 1). The majority of patients were discharged home (80.95%), and only a small proportion were admitted to ICU (7.48%). The overall 30-day mortality rate was 24 (16.33%) (Table 1), comprising 15 (62.5%) in-hospital deaths and 9 (37.5%) post-discharge deaths.

### 2.2. Guideline Adherence

Amoxicillin–clavulanic acid was the most widely used antibiotic therapy, administered to 29.07% of patients in monotherapy and 54.09% of patients in combination, followed by ceftriaxone (monotherapy: 29.07%, combination: 25.59%) and moxifloxacin (monotherapy: 19.77%, combination: 16.39%) (Table 2). Guideline adherence (agent choice) rates to national, BTS/NICE, and ATS/IDSA CAP guidelines are presented in Table 3. Initial empirical therapies for CAP showed a relatively low rate of guideline adherence: 30.61% for national, 22.45% for BTS/NICE, and 15.65% for ATS/IDSA CAP guidelines. The rate of adherence to at least one guideline was 34.69% (Table 3).

Dosage appropriateness assessments are shown in Table 4. In line with the previous section, the highest guideline adherence (agent, dose) rate was found in relation to the national guideline (40/45, 88.89%), followed by ATS/IDSA (18/23, 78.26%) and BTS/NICE (24/33, 72.73%) CAP guidelines.

### 2.3. Antibiotic Therapy for CAP

The characteristics of first antibiotic therapies and key outcomes are described in Table 3. The majority of treatments (58.50%) were monotherapies; 93 (63.27%) patients received the first antibacterial therapy IV (intravenously), and 14 of them (15.05%) were switched to oral route within 1–5 (median 3.5) days.

The average duration of antibiotic therapy for CAP was 7.13 ± 4.37 days (median 6, range 1–27), while the average antibiotic consumption was 11.41 ± 8.59 DDD/patient (range 1–44.5). The majority of patients (81.63%) received short-term (1–6 days) antibiotic therapy. In the majority of cases, there was no change in the first empirical therapy (85/147, 57.8%). However, changes occurred due to sequential antibiotic therapy (9.52%), de-escalation (4.08%), and escalation (28.57%) (Table 3). A significant difference was found in the 30-day mortality rate between these types of antibiotic therapies (no change: 12.94%, sequential antibiotic therapy: 0%, de-escalation: 0%, and escalation: 30.95%, *p* = 0.046).

### 2.4. Clinical Outcomes: LOS, 30-Day Mortality

In our study, the mean LOS was 8.26 ± 5.64 (range 1–33) days (Table 1). Adherence to the national guideline led to a slightly lower 30-day mortality rate than guideline non-adherence (15.56% vs. 16.67%, *p* > 0.05), while this difference was more pronounced in the case of international guidelines (BTS/NICE: 21.21% vs. 14.91%, and ATS/IDSA: 21.74 vs. 15.32%, *p* > 0.05) (Table 3). Furthermore, we found that the 30-day mortality rate for the different types of therapies was as follows: 8% for combination of beta-lactam and macrolide, 19.61% for beta-lactam monotherapies, and 21.77% for respiratory fluoroquinolone monotherapies (*p* > 0.05).

### 2.5. Prognostic Factors for Mortality in CAP

The demographic and clinical characteristics of 30-day survivors (123/147, 83.67%) and non-survivors are compared in Table 5.

We observed a significant difference in the 30-day mortality of CAP between age groups. The 30-day mortality rate increased proportionally with age: it was 6.90% (2/29) among patients aged 20–64 years, 11.11% (8/72) in patients aged 65–84 years, and reached 30.43% (14/46) in the 85+ age group (Table 5).

The CCI score of patients in the 30-day non-survivor group was higher by one point on average (5.71 ± 1.85 vs. 4.67 ± 1.83, *p* = 0.012) (Table 5).

In terms of C-reactive protein (CRP) levels at admission, a remarkable difference was found between the two patient groups (30-day non-survivor: 177.28 ± 118.94 vs. 30-day survivor: 112.88 ± 93.47 mg/L, *p* = 0.006) (Table 5).

Thirty-day mortality was not associated with significantly longer LOS (9.54 ± 8.45 vs. 8.01 ± 4.93 days, *p* = 0.668), higher antibiotic exposure (8.25 vs. 7.98 DDD/patient, *p* = 0.21), or longer duration of antibiotic therapy (8.20 ± 7.03 vs. 6.92 ± 3.64 days, *p* = 0.187). Similarly, we found a median 1-day difference between 30-day survivors and non-survivors in the duration of antibiotic therapies (6 vs. 7 days, respectively), and length of stay (7 vs. 8 days, respectively).

The results of logistic regression analysis are displayed in Table 6. Out of the three factors (increased age, higher CCI score, and higher CRP level) that were associated with higher mortality in the univariate analysis, only the CRP level on admission was found to increase the risk of mortality. Each additional increase of 50 mg/L in the CRP level seen on admission increased the 30-day mortality odds 1.3-fold, indicating that the degree of inflammation affects mortality.

## 3. Discussion

Even though CAP is one of the most common acute infections, ours is the first field study in Hungary that has been conducted regarding the evaluation of antibiotic prescription patterns, associations between guideline adherence and outcomes in patients with CAP who required hospitalization.

### 3.1. CAP Guidelines

Based on ATS/IDSA and BTS/NICE *CAP* guidelines, combinations of beta-lactams and macrolides, or respiratory fluoroquinolones (RFQs) are recommended as first choice agents to treat empirically moderate-severe (hospitalized in non-ICU ward) CAP [19,20].

The Hungarian guideline for patients hospitalized with CAP is similar to international guidelines in terms of agent selection [21]. This guideline recommends the use of respiratory fluoroquinolones (moxifloxacin or levofloxacin) as monotherapy or the combination of beta-lactam (amoxicillin clavulanic acid or ceftriaxone) and clarithromycin to cover both typical (e.g., *Streptococcus pneumoniae*, *Haemophilus influenzae*, *Staphylococcus aureus*, Group A streptococci, *Moraxella catarrhalis*) and atypical pathogens (e.g., *Legionella*, *Mycoplasma pneumoniae*, *Chlamydia pneumoniae*) responsible for CAP.

#### 3.1.1. Guideline Adherence: Agent Selection

Among the patients hospitalized with CAP investigated in the present study, the rate of national guideline adherence for antibiotic selection was 30.61% (N = 45). The most common guideline adherent empirical treatment for CAP was amoxicillin-clavulanic acid combined with clarithromycin, or moxifloxacin or levofloxacin as monotherapy (23, 47.92% in both cases), followed by ceftriaxone combined with clarithromycin (2, 4.16%). In 2017, national surveillance data for antibiotic resistance in hospitalized patients still reported relatively high susceptibility rates for the antibacterial agents used against *S. pneumonia* (98.5% to ceftriaxone, 96.3% to levofloxacin, 96.2% to moxifloxacin 93.8% to ampicillin, and 74.7% to macrolides). Additionally, amoxicillin clavulanic acid showed potent activity (94.4%) against *H. influenza* strains [22].

Guideline adherent empirical antibiotic use in CAP is quite varied in the related literature. Three studies evaluating patients hospitalized with CAP found guideline adherent antibiotic therapy in 57%, 57%, and 65% of the cases [16,17,23]; these rates were higher compared to our results. At the same time, an Italian multicenter before-and-after guideline implementation survey found that guideline adherent antibiotic prescribing increased significantly (33 vs. 44 %; *p* < 0.001) [24] compared to a poor initial guideline adherence, similar to our results. The low guideline adherence found in our study may be explained by the fact that although there was a Hungarian guideline, its dissemination and accessibility were not adequate; consequently, it had not been integrated in daily practice.

#### 3.1.2. Guideline Adherence: Dosing

Even though we found high adherence to the national guideline in terms of dosing (88.89%), over- and underdosing still affected relatively high proportions of patients (8.89% and 2.22%, respectively). Overdosing occurred most commonly in renal impairment, when dose adjustment would have been required for amoxicillin-clavulanic acid, clarithromycin, and moxifloxacin. The other common error occurred mostly due to routine underdosing of levofloxacin and clarithromycin, or not taking into account patients’ extreme body weights (Appendix A: Table A1).

### 3.2. Changes in the First Empirical Therapy

Considering the route of administration, the majority of patients (63.27%) received IV initial antibacterial therapy for CAP. At the same time, switching from an IV to oral regimen (in 9.52% of the cases) was performed within 1–5 (median 3.5) days. These results are mostly supported by the national and international guidelines, according to which the empirical antibiotic treatment in patients hospitalized with CAP can be initiated via any route, but using antibiotics exclusively intravenously is only recommended when the oral route is compromised. The review of intravenous antibiotics after 48 h of use and switching to oral antibiotics are recommended, if possible, when either the same agent or the same drug class should be used [19,20]. According to the ATS/IDSA guideline, patients hospitalized with CAP should be switched from intravenous to oral therapy when they are hemodynamically stable, showing signs of clinical improvement (within the first 48–72 h), are able to ingest medications, and have a normally functioning gastrointestinal tract [25]. At the same time, according to a multicenter randomized clinical trial performed in four teaching hospitals in Spain, the switch from intravenous to oral regimen is not currently common in clinical practice [26].

In addition, more antibiotic therapy needed further escalation (28.57%), while changes in the first empirical therapy due to de-escalation (4.08%) occurred at relatively low rates.

The guidelines for CAP stress the importance of de-escalation of empirical antibiotic therapy, recommending the stricter use of broad-spectrum antibiotics [19,20]. Although appropriate dosage and de-escalation are important in optimizing antibiotic use and reducing antibiotic resistance, studies dealing with antibiotic dosing in CAP treatments are rare. A cross-sectional study in Australian patients hospitalized with CAP found that the most common errors in high-risk CAP were inappropriate dose, route, and duration, which affected 69% (N = 27) of patients. Routine underdosing of ceftriaxone was the most frequent (N = 17, 44%), while 54% of patients were prescribed antibiotics to administer via a route not recommended on the basis of CAP severity [27]. According to a multicenter study in the Netherlands, where de-escalation occurred in 16.7% of the patients hospitalized with CAP, physicians seem to be more inclined to continue the regimen when it appears to be effective [28].

### 3.3. Duration of Antibiotic Therapy

Our results are in line with the requirements of international guidelines [19,20]: most of our patients (81.63%) receive short antibiotic therapy (1–6 days), while the median duration of antibiotic therapies for CAP was 6 days (range 1–27).

The optimal duration of antimicrobial therapy in CAP is not well-established. Although the national CAP guideline for in-patients does not cover the duration of antibiotic treatment, according to the ATS/IDSA guideline, patients hospitalized with CAP should be treated for a minimum of 5 days [25]. Additionally, in inpatient settings, a small number of studies have addressed the appropriate duration of antibiotic therapy in CAP. A recent meta-analysis of patients hospitalized with CAP demonstrated the efficacy of shorter courses of antibiotic therapy (of 5 to 7 days) [29]. Despite recommendations, a recent international audit found that prolonged antibiotic therapy for CAP was common and frequently observed due to the presence of comorbidities [30].

### 3.4. Clinical Outcomes: 30-Day Mortality

Regarding clinical outcomes, in the present study we found that guideline adherence to national recommendations was associated with slightly lower 30-day mortality than guideline non-adherence (15.56% vs. 16.67%, *p* > 0.05). Furthermore, studies showed that both in Europe and the United States, guideline adherence in patients hospitalized with CAP was associated with lower 30-day mortality [13,14,15]. Nevertheless, another multicenter cross-sectional study reported that no significant difference was found between guideline adherent and non-adherent antibiotic prescribing episodes and inpatient mortality (1.6% vs. 4.1%; *p* = 0.18) [31].

Several studies have focused on the relation between mono- or combination therapies and clinical outcomes [32,33]. The results of a multicenter study in patients admitted to non-ICU wards with CAP have shown clinical outcomes, recovery rate and mortality to be unaffected by the choice of a beta-lactam, beta-lactam and macrolide, or respiratory fluoroquinolone antibiotic regimen [32]. According to a systematic review on antibiotic therapy for non-ICU hospitalized patients with CAP, fluoroquinolone monotherapy had similar efficacy and favorable safety compared to beta-lactam with or without macrolide [34]; however, the authors pointed out several quality issues and recommended further good quality research to confirm these findings [34].

In the present study, we found a slightly better mortality rate in CAP hospitalized patients with the combination of beta-lactam and macrolide, compared with beta-lactam or respiratory fluoroquinolone monotherapies (8% vs. 19.61% and 21.77%, respectively, *p* > 0.05).

Further, changes in the first empirical therapy due to de-escalation (4.08%) and switching from intravenous to oral regimen (9.52%) occurred relatively infrequently, and were not associated with increased 30-day mortality rates (0% for both). Admittedly, we conducted the survey on a relatively small number of cases. A simulation study embedded in a prospective cohort (performed in 58 hospitals) found that 30-day mortality in patients hospitalized with CAP was 3.5% and 10.9% in the de-escalation and continuation groups, respectively. At the same time, the simulation study also suggested that the effect of de-escalation on mortality needs further evaluation to determine effect size more accurately [28].

Regarding the duration of antibiotic therapy, we found no difference in mortality rates between short- and long-term therapies (16.67% vs. 14.81%, p>0.05), which may suggest that short antibiotic therapy can be as effective as long antibiotic therapy. A previous meta-analysis of five randomized trials (which included patients of all ages, excluded neonates, and any severity of CAP) found no differences in clinical outcome and mortality rates comparing short (1–6 days) versus long (≥7 days) antibacterial therapies [35]. Our results support these finding by showing similar mortality rates for both short and long antibiotic durations.

### 3.5. Prognostic Factors for Mortality Due to CAP

Previous research found that increased age, male gender, increased CRP, and comorbid conditions (mainly malignancy, congestive heart failure, diabetes mellitus, and renal disease) act as predictive factors for mortality in patients hospitalized with CAP [36,37,38].

As for age, our results show that 30-day mortality in patients aged ≥ 85 years was 3-fold compared with those aged 65–84 years (30.43% vs. 11.11%). Studies found that age ≥ 85 years was an independent predictive factor for mortality in CAP, increasing the risk of death significantly [36,37]. According to Torner et al., age ≥ 85 years was markedly associated with mortality in CAP, since the 30-day mortality rate was 2.6 times higher in this age group compared with patients aged between 65 and 84 years [39]. Moreover, Luna et al. concluded that an age of 80 years or more should already be considered a risk factor for poor outcome in CAP [40].

Furthermore, a temporal analysis of pneumonia (excluding influenza-related pneumonia, aspirational pneumonia, and congenital pneumonia) mortality rates in European countries between 2001 and 2014 revealed gender discrepancy: mortality was higher in males than in females [41]. Regarding Hungary, a mortality rate of 7.46% in males and 3.72% in females was reported [41]. Surprisingly, the mortality rate in the present study was higher among females than males (18.07 vs. 14.06%). However, this difference is not clinically significant. Even though in the study population there were more females (56.46%) than males, we cannot give an obvious explanation for these mortality rates, since CCI and CRP did not differ across genders.

The other commonly studied prognostic factor for CAP mortality is CRP level. The CRP test is the most widely used serum biomarker in the differential diagnosis (viral or bacterial etiology) of lower respiratory tract infections. Due to bacterial infection, CRP levels rise within the first 6 to 8 h in response to several inflammatory stimuli.

Several studies evaluated the relationship between C-reactive protein serum level and outcomes of CAP. Mendez et al. and Summah et al. concluded that CRP values increase in line with the severity of CAP, and can be used as an independent prognostic predictor of the severity of CAP, for the follow-up of patients’ condition, for response to antibiotic therapy, and CAP clinical outcome [42,43,44,45]. Moreover, CRP level may guide CAP empirical treatment decisions and help avoid unnecessary antibiotic use in hospitalized patients [46,47]. A recent study conducted in a Scottish hospital demonstrated that a CRP level below 100 mg/L on admission was significantly associated with reduced 30-day mortality (OR 0.18, *p* = 0.03) [48]. In a Danish teaching hospital, the highest mortality risk was found in patients with CRP > 75 mg/L on admission [49]. Results of the present study are consistent with these previous findings, as we recorded significantly higher average CRP values on admission in the group of patients who died within 30 days compared to 30-day survivors (177.28 ± 118.94 vs. 112.88 ± 93.47 mg/L, *p* = 0.006).

Regarding comorbid conditions, we found that CCI scores differed significantly between the 30-day non-surviving and 30-day surviving patients (5.71 ± 1.85 and 4.67 ± 1.83, *p* = 0.012). A higher CCI score due to the presence of comorbidities was associated with higher mortality rates (CCI score 0–4: 11.86%, CCI score 5–10: 19.32%) in CAP, similar to other literature data. A secondary analysis of CAP performed by Luna et al. found that the presence of comorbidities was associated with poorer outcomes [40].

### 3.6. Strengths and Limitations

The collected data provide detailed, first-hand observations on the everyday use of antibiotics in the empirical treatment of CAP in internal medicine hospitals. However, retrospective data collection from medical records might contain inaccuracies and potential biases.

One of the most important limitations of this study was that no clinical case definition of CAP was given or standardized at hospital level. However, the diagnosis of pneumonia was confirmed in every case by chest radiography. The second limitation was the lack of knowledge of pneumonia severity score (PSI score), since not all elements of the score were retrievable from medical records. Furthermore, there were no set hospital standard guidelines for the empirical antibiotic treatment of CAP. Therefore, national and international guidelines were used for assessing antibiotic use. Third, we also consider it likely that de-escalation (prescribing an oral antibiotic) occurred after discharge. However, no data were collected on de-escalation after discharge.

In conclusion, this study provides further evidence that guideline adherence in choosing the empirical antibiotic improves survival, and thus contributes to improvement of acceptance of antimicrobial stewardship. The results also draw attention to the need for improvement of empirical prescribing by limiting unnecessary combinations and by optimizing doses, especially in the cases of patients with higher CRP, In our country, there are few studies that explore those important healthcare practices at the individual patient level that may lead to the development of antimicrobial resistance. We believe that our results may contribute to optimizing CAP treatment in the future.

## 4. Materials and Methods

### 4.1. Study Design and Setting

A 1-year (January–December 2017) retrospective observational study was conducted at the 110-bed internal medicine unit of the University of Debrecen, which is a tertiary care teaching hospital.

### 4.2. Data Collection

Data for all inpatients receiving antibacterial therapy during the hospital stay were recorded by the ward pharmacist. All patient and therapy related data were collected manually from medication charts and discharge letters using the e-MedSolution Hospital Information System. Data collection forms were developed and the following data were extracted: patient age, sex, weight, date of hospital admission and discharge, comorbidities, discharge type. Clinical outcome (30-day mortality) and laboratory test results on the day of admission (white blood cell count, CRP, eGFR—estimated glomerular filtration rate) were also collected. In relation to the antibacterial therapy, the following data were collected: pre-hospital antibiotic therapy, drug allergy, indication of antibiotic treatment, empirical antibiotic choice, dosage, route of administration, and duration of antibacterial therapy during hospital stay. The extracted data were entered into Microsoft Excel spreadsheets for further analysis.

Only adult (18 years or above) patients who started their first empirical antibacterial therapy for community acquired pneumonia were included in the study. Empirical treatment was defined as antibacterial therapy without pathogen identification and susceptibility testing. Inclusion and exclusion criteria for the study are shown in Figure 1.

Patients’ general condition was evaluated using the Charlson comorbidity index (CCI) [50]. eGFR on admission was used to assess dose appropriateness for drugs excreted renally. To reveal the antibiotic exposure of patients, the World Health Organization’s ATC/DDD index (version 2021) was applied. Defined daily dose (DDD) refers to the assumed average maintenance dose per day for a drug used for its main indication in adults. Regarding antibiotics, DDD refers to infections of moderate severity [51]. Our analysis focused on systemic antibacterial drugs (ATC: J01). LOS refers to the number of days that patients spent in hospital. Both the admission and discharge day were counted as a separate day.

### 4.3. Main Outcome Measures

The primary outcome measure was guideline adherence to the national (published by Hungarian Professional College of Infectious Diseases and Pulmonology) and two international (ATS/IDSA-American Thoracic Society/Infectious Diseases Society of America, BTS/NICE-British Thoracic Society/National Institute for Health *and* Care Excellence) CAP guidelines, in terms of choice of empirical antibiotic(s) and dosing. Therefore, empirical treatment was considered guideline adherent when complying with the recommendations.

Secondary outcome measures included antibiotic exposure (DDD/patient), and clinical outcome (30-day mortality rate).

Furthermore, demographic (age, gender) and clinical characteristics (CCI, CRP) of patients with CAP in the 30-day mortality and 30-day survivor groups were compared.

Assessment for guideline adherence was performed separately for each guideline as follows:

*Choice assessment:* The first empiric antibiotic therapy initiated for patients hospitalized with CAP was matched with guideline recommendations on antibiotic choice, and classified as adherent or non-adherent. Combined therapy was considered guideline adherent when all antibacterial agents of the combination were adherent. Non-immunocompetent patients (malignancy) were excluded from guideline adherence analysis, as the guidelines did not cover this special population.

*Dosage assessment:* The dose of the first guideline adherent empiric antibiotic therapy was established on the basis of the guidelines mentioned above, and defined as follows:-*appropriate dose:* dose recommended by guidelines, administration of loading dose when recommended, and dose adjustment in renal impairment.-*debatable dose:* under- or overdose by <50% compared to the dose recommended by guidelines, and/or absence of loading dose.-*under-or overdose*: under- or overdose by ≥50% compared to the dose recommended by guidelines, and/or no dose adjustment in renal impairment and in extremes for body weight.

In cases of extreme body weight (<40 and >100 kg) and impaired renal function, the summary of product characteristics (SPC) was also considered, as it gives a detailed description about how to take into account body weight and eGFR in dose calculation. Dosing assessment was not performed for therapies considered as non-adherent regarding the antibiotic choice.

Changes in the first antibacterial treatment (sequential therapy: switch from an IV to oral regimen, de-escalation or escalation) were also assessed. Narrowing spectrum was considered de-escalation, while adding a new antibiotic or switching to a broader-spectrum agent was defined as escalation of the antibiotic regimen.

*Clinical outcome assessment:* The clinical outcome assessment was performed to see whether adherence to CAP guidelines improved 30-day mortality, and to map the predictive factors for mortality in patients hospitalized with CAP.

### 4.4. Statistical Analyses

Quantile–quantile plots (Q–Q plots) and density plots were used for checking normality of data visually. Fisher’s exact test was applied to compare categorical variables, and t-test was used to compare continuous variables between groups. Significant p values were defined as below 0.05.

Patients were anonymized, thus made unidentifiable in the study.

## 5. Conclusions

We found poor adherence to the national and international CAP guidelines in terms of agent choice. In addition, CRP value on admission was markedly associated with mortality in CAP.

## Figures and Tables

**Figure 1 antibiotics-11-00468-f001:**
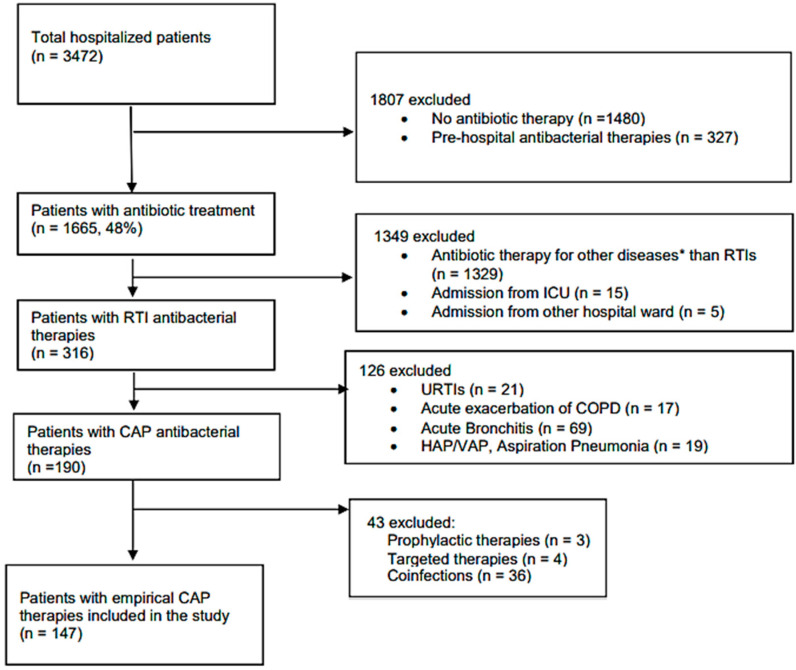
Flowchart for exclusion and inclusion criteria. * Other diseases: e.g., sepsis, urinary tract infection, etc.; RTIs—respiratory tract infections; ICU—intensive care unit; URTIs—upper respiratory tract infections; COPD—chronic obstructive pulmonary diseases; HAP—hospital acquired pneumonia; VAP—ventilation associated pneumonia; CAP—community acquired pneumonia.

**Table 1 antibiotics-11-00468-t001:** Demographic and clinical characteristics of patients with CAP.

Parameter	N	%
	147	100
Gender (Male)	64	43.54
Age		
20–64 years	29	19.73
65–84 years	72	48.98
≥85 years	46	31.29
Penicillin allergy	2	1.36
CCI—Charlson comorbidity index		
0	3	2.04
1	2	1.36
2	10	6.80
3	12	8.16
4	32	21.77
>4	88	59.86
Comorbidities		
Cardiovascular disease	52	35.37
Diabetes mellitus	33	22.45
Chronic obstructive pulmonary disease	13	8.84
Chronic liver/kidney disease (moderate to severe)	11	7.48
Hematologic malignant diseases	8	5.44
Solid tumor		
Localized	2	1.36
Metastatic	6	4.08
Peripheral vascular disease	5	3.40
Dementia	3	2.04
Peptic ulcer disease	2	1.36
Cerebrovascular accident or transient ischemic attack	1	0.68
Discharge types		
Discharged home	119	80.95
Moved to another hospital ward	2	1.36
Intensive care unit (ICU)	11	7.48
Outcome		
In-hospital mortality	15	10.20
30-day mortality	24	16.33
Length of stay (LOS) (mean ± SD)-days	8.26 ± 5.64 (1–33) *	

SD–standard deviation; * Data are presented as mean ± standard deviation (min–max).

**Table 2 antibiotics-11-00468-t002:** The distribution of first empirical antibiotic therapies (mono- and combination therapies).

Antibiotics	Frequency (N)		%	Guideline Adherence		
				National	BTS/NICE/NICE	ATS/IDSA
**Monotherapies (N = 86; 100%)**						
Amoxicillin-clavulanic acid	25		29.07			
Ceftriaxone	25		29.07			
Moxifloxacin	17		19.77	✓		✓
Levofloxacin	6		6.98	✓	✓	✓
Clarithromycin	5		5.81		✓	
Meropenem	4		4.65			
Amoxicillin	1		1.16		✓	
Doxycycline	1		1.16		✓	
Metronidazole	1		1.16			
Norfloxacin	1		1.16			
**Combination therapies (N = 61; 100%)**						
amoxicillin-clavulanic acid + clarithromycin	23	37.70		✓	✓	
moxifloxacin + metronidazole	7	11.48				
ceftriaxone + metronidazole	6	9.84				
amoxicillin-clavulanic acid + metronidazole	5	8.20				
amoxicillin-clavulanic acid + clarithromycin + metronidazole	3	4.92				
ceftriaxone + clarithromycin	2	3.28		✓		✓
ceftriaxone + metronidazole + clarithromycin	2	3.28				
ceftriaxone + sulphamethoxazole and trimethoprim	2	3.28				
amoxicillin-clavulanic acid + clarithromycin + amikacin	1	1.64				
amoxicillin-clavulanic acid + flucloxacillin	1	1.64				
ceftriaxone + metronidazole + sulphamethoxazole and trimethoprim	1	1.64				
ceftriaxone + moxifloxacin	1	1.64				✓
levofloxacin + metronidazole	1	1.64				
meropenem + metronidazole	1	1.64				
moxifloxacin + flucloxacillin	1	1.64				
moxifloxacin + metronidazole + ceftriaxone	1	1.64				
piperacillin/tazobactame + amikacin	1	1.64				
piperacillin/tazobactame + metronidazole	1	1.64				
meropenem + vancomycin	1	1.64				

ATS/IDSA—American Thoracic Society/Infectious Diseases Society of America; BTS/NICE—British Thoracic Society/National Institute for Health and Care Excellence; ✓—guideline adherence.

**Table 3 antibiotics-11-00468-t003:** Characteristics of antibiotic therapies.

Parameters	N	%
	147	100
Adherence to the national guideline (agent choice)	45	30.61
Adherence to BTS/NICE guideline (agent choice)	33	22.45
Adherence to ATS/IDSA guideline (agent choice)	23	15.65
Adherence to at least one guideline (agent choice)	51	34.69
Type of the first antibiotic therapy		
Combination therapies	61	41.50
Monotherapies	86	58.50
Most common therapies		
beta-lactams and macrolide	25	17.01
beta-lactams	51	34.69
respiratory fluoroquinolones	23	15.65
Route of administration of the first antibiotic therapy		
iv	93	63.27
oral	54	36.73
Duration of total antibiotic therapies		
short therapy (1–6 days)	120	81.63
long therapy (≥ 7 days)	27	18.37
Number of consecutive antibiotic therapies		
1	85	57.8
>1 (2–4)	62	42.2
Changes in the first empirical therapy		
Sequential antibiotic therapy*	14	9.52
De-escalation	6	4.08
Escalation	42	28.57
No change	85	57.8

BTS/NICE—British Thoracic Society/National Institute for Health and Care Excellence; ATS/IDSA—American Thoracic Society/Infectious Diseases Society of America; iv—intravenously; * switch from an IV to oral regimen.

**Table 4 antibiotics-11-00468-t004:** Guideline adherence, N = 147 patients.

	Adherence Frequency	%
**AB1-National CAP guideline adherence**	**45**	100
appropriate use	40	88.89
overdose (compared to SPC, due to lack of guideline recommended dose)	4	8.89
underdose (due to body weight)	1	2.22
**AB1-BTS/NICE CAP guideline adherence**	**33**	100
appropriate use	24	72.73
underdose (compared to guideline)	4	12.12
overdose (in case of low levels of eGFR)	4	12.12
debatable use (absence of loading dose)	1	3.03
**AB1-ATS/IDSA CAP guideline adherence**	**23**	100
appropriate use	18	78.26
underdose (compared to guideline)	3	13.04
overdose (in case of low levels of eGFR)	2	8.70

AB1—first empirical antibiotic treatment; CAP—community acquired pneumonia; SPC—summary of product characteristics; BTS/NICE—British Thoracic Society/National Institute for Health and Care Excellence; eGFR—estimated glomerular filtration rate; ATS/IDSA—American Thoracic Society/Infectious Diseases Society of America.

**Table 5 antibiotics-11-00468-t005:** Comparison of baseline and clinical characteristics of 30-day non-survivors and survivors among patients with CAP.

		30-Day Survival		
		Non-Survivors	Survivors	*p*-Value
Total		24 (16.33%)	123 (83.67%)	**-**
Gender	male	9 (14.06%)	55 (85.94%)	0.654
	female	15 (18.07%)	68 (81.93%)	
Age (years)	mean ± SD	81.57 ± 10.77	75.12 ± 13.43	0.028
	20–64	2 (6.90%)	27 (93.1%)	-
	65–84	8 (11.11%)	64 (88.89%)	
	85+	14 (30.43%)	32 (69.57%)	
CCI score	mean ± SD	5.71 ± 1.85	4.67 ± 1.83	0.012
Diabetes mellitus	yes	7 (21.21%)	26 (78.79%)	0.425
	no	17 (14.91%)	97 (85.09%)	
Leukemia	yes	5 (13.89%)	31 (86.11%)	0.798
	no	19 (17.12%)	92 (82.88%)	
Chronic kidney disease	yes	4 (57.14%)	3 (42.86%)	0.014
	no	20 (14.29%)	120 (85.71%)	
Congestive heart failure	yes	6 (12.77%)	41 (87.23%)	0.482
	no	18 (18%)	82 (82%)	
Type of therapy	combination	6 (9.84%)	55 (90.16%)	0.112
	monotherapy	18 (20.93%)	68 (79.07%)	
National CAP guideline adherence	adherent	7 (15.56%)	38 (84.44%)	1.000
	non-adherent	17 (16.67%)	85 (83.33%)	
BTS/NICE CAP guideline adherence	adherent	7 (21.21%)	26 (78.79%)	0.425
	non-adherent	17 (14.91%)	97 (85.09%)	
ATS/IDSA CAP guideline adherence	adherent	5 (21.74%)	18 (78.26%)	0.538
	non-adherent	19 (15.32%)	105 (84.68%)	
CRP (mg/L) at admission	mean ± SD	177.28 ± 118.94	112.88 ± 93.47	0.006
	high levels (8˂)	20 (16.67%)	101 (83.47%)	0.449
	normal levels (0–8)	1 (8.33%)	10 (90.91%)	
	NA	3 (20%)	12 (80%)	-

SD—standard deviation; CCI—Charlson comorbidity index; CAP—community acquired pneumonia; BTS/NICE-British Thoracic Society/National Institute for Health and Care Excellence; ATS/IDSA—American Thoracic Society/Infectious Diseases Society of America; CRP—C-reactive protein; NA—not available. *p*-value: Fisher’s exact test was performed for categorical variables, and *t*-test was used to compare continuous variables between groups.

**Table 6 antibiotics-11-00468-t006:** CCI scores and CRP levels on admission in non-surviving and surviving patients’ groups with odds ratio.

	B	S.E.	*p*-Value	OR	95% CI for OR	
					Lower	Upper
Age (years)	0.058	0.032	0.072	1.059	0.995	1.128
CCI score	0.203	0.155	0.191	1.2259	0.904	1.659
CRP 9 category *	0.289	0.125	0.020	1.3362	1.046	1.705
Constant	−8.562	2.675	0.001	0.000		

B—regression coefficient; S.E.—standard error; OR—odds ratio; CI—confidence interval; CCI—Charlson comorbidity index; CRP—C-reactive protein; * CRP 9 categories: 1: 0–8 (mg/L); 2: 8–50 (mg/L); 3: 50–100 (mg/L); 4: 100–150 (mg/L); 5: 150–200 (mg/L); 6: 200–250 (mg/L); 7: 250–300 (mg/L); 8: 300–350 (mg/L); 9: above 350 (mg/L).

## Data Availability

Data are available from the corresponding author upon reasonable request.

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
