# Peer review of "Impact of Guideline Adherence on Outcomes in Patients Hospitalized with Community-Acquired Pneumonia (CAP) in Hungary: A Retrospective Observational Study"

_antibiotics, 2022, doi:10.3390/antibiotics11040468_

Round 1

Reviewer 1 Report

Interesting work but few clarities are required;

  1. Why national & International 2/3 types of guidelines were taken as a standard?
  2. It is suggested to provide information on these guidelines in the introduction, as appendix A is indicated the names only.
  3. In the Abstract, the author mentioned that…” The main outcome measures were adherence rate to national and…”, Adherence rate might not be a rational term. As the rate of adherence didn’t mention in the conclusion. Rephrasing is suggested.
  4. Abbreviations shouldn’t be placed prior to the full text.
  5. The Methodology should be placed earlier than the Result and Discussion section.
  6. The methodology section should be in detail enough and self-explanatory.
  7. Why only RTI was selected? Any rationale for that.
  8. Any reason that comorbidities were excluded with reference to figure 1 (Flow chart of the study)?, while the table1 indicated the comorbidities?
  9. Why cited 3 guidelines were considered as a reference. Are these guidelines being exactly the same recommendation to treat CAP or variations are existing? If YES then why 3 guidelines considered as a standard, why not only 1. If NO, then the reliability of the outcome needs to be re-look. I will suggest overcoming this point by adding a detailed introduction and criteria.
  10. Data collection tool validity needs to incorporate in detail.
  11. The sample size is another concern.
  12. Ethical approval from any source and its number need to clarify.
  13. The conclusion should be based on the study hypothesis or objectives and rephrasing is advised. Abstract conclusion and general conclusion should be more clear.
  14. Do we generalize this picture with the whole country's perspective?
  15. The present shape looks like a dissertation/thesis, the author needs to convert it into a proper research article format.

Author Response

Response to Reviewer 1.

We thank the Reviewer for the helpful comments to which we reply as follows.

The changes are highlighted in the manuscript with track changes.

Comment 1: Why national & International 2/3 types of guidelines were taken as a standard?

Response: We wished to compare our clinical practise to our national guideline, as well as with current international standards. The choice fell on these guidelines because they are the best known and most easily accessible by Hungarian physicians. In the absence of local CAP guideline, these can guide physicians. This kind of double, triple comparison is used in the scientific literature (e.g. PMID: 22338542 or PMID: 33602106 or PMID: 26046802) and add to the evaluation.

Comment 2: It is suggested to provide information on these guidelines in the introduction, as appendix A is indicated the names only.

Response: We added this information as suggested.

Comment 3: In the Abstract, the author mentioned that…” The main outcome measures were adherence rate to national and…”, Adherence rate might not be a rational term. As the rate of adherence didn’t mention in the conclusion. Rephrasing is suggested.

Response: The term “adherence rate” was corrected to guideline adherence.

Comment 4: Abbreviations shouldn’t be placed prior to the full text.

Response: The necessary corrections were made.

Comment 5: The Methodology should be placed earlier than the Result and Discussion section.

Response: Placing methods after results and discussion was a requirement of the journal.

Comment 6: The methodology section should be in detail enough and self-explanatory.

Response: The methods section is divided into four different subchapters where we clearly reported study design, setting, way of data collection, exclusion criteria, calculation of outcome measures and used statistics. We added a flow chart for easier understanding of patients included.

Comment 7: Why only RTI was selected? Any rationale for that.

Response: As we phrased in the introduction CAP is one of the most important infections with high burden. We tried to achieve as homogenous patient population as possible in order to enable in depth, meaningful analysis, so we excluded other RTIs than CAP. In the future we also aim to analyse antibiotic use in Urinary Tract Infections (UTIs), the other most common encountered infection.

Comment 8: Any reason that comorbidities were excluded with reference to figure 1 (Flow chart of the study)?, while the table1 indicated the comorbidities?

Response: On Figure 1 (Flow chart of the study) we listed exclusion and inclusion criteria. Patients with other type of respiratory infections were excluded (see my comment above). If patient had another type of infection beside CAP, we also excluded them, as there is no means to evaluate CAP guideline adherence in these cases. In Table 1. we indicated common comorbidities of the included patients (which did not require antibiotic therapy) and not the coinfections.

Comment 9: Why cited 3 guidelines were considered as a reference. Are these guidelines being exactly the same recommendation to treat CAP or variations are existing? If YES then why 3 guidelines considered as a standard, why not only 1. If NO, then the reliability of the outcome needs to be re-look. I will suggest overcoming this point by adding a detailed introduction and criteria.

Response: Please see also our comment above (answer to the first comment). Of course the Hungarian guideline for patients hospitalized with CAP is similar to international guidelines in terms of agent selection (Figure A1), but there are some differences. The differences are mainly due to the classification of the recommended agents as first- or second-line agents (e.g. according to the BTS/NICE guideline respiratory fluoroquinolones are second-line agents, while according to the Hungarian guideline are first-line agents), doses (e.g. levofloxacin, see Figure A1), and the use of different active agents within the same class of antibiotics.

Comment 10: Data collection tool validity needs to incorporate in detail.

Response: We did not use any data collection tool, but a continuous, manual chart review was performed. The data collection was performed by the first author, which provide consistency throughout the study.

Comment 11: The sample size is another concern.

Response:  Sample size calculation was not performed in advance due to the various outcome measures. However, if we consider one of the main clinical outcomes, the mortality, as its incidence was between 10% and 90% (it was 16,33 %, so not so rare and not so frequent) the sample size needed should be between 138 and 384 (see e.g. Ravindra Arya et al. Sample Size Estimation in Prevalence Studies Indian J Pediatr 79(11):1482 – 1488. So our sample fulfilled this requirement, we reached the minimal threshold. Enrolling more patient could have been achieved only if use less stringent exclusion criteria, which would result in less homogenous patient cohort.

Comment 12: Ethical approval from any source and its number need to clarify.

Response: The study was conducted before the introduction of GDPR: The study was performed in accordance with the local ethical and legal requirements. According to Hungarian legislation, ethical approval and written informed consent was not required as the study served for quality development, and the study and relied on the Summary of Product Characteristics. Furthermore, participating in ASP and data collection was part of the pharmacist’s daily routine.

Comment 13: The conclusion should be based on the study hypothesis or objectives and rephrasing is advised. Abstract conclusion and general conclusion should be more clear.

Response: Corrections were made as suggested.

Comment 14: Do we generalize this picture with the whole country's perspective?

Response: This study, as all single-center study, so the reported findings are not directly generalizable, as stated in the limitations. However, as we detected this suboptimal practice in a university affiliated hospital (highest level of care), we might suppose that other hospitals have similar problems.

Comment 15: The present shape looks like a dissertation/thesis, the author needs to convert it into a proper research article format.

Response: The present “shape” was a requirement of the journal.

We hope that these changes have improved the manuscript.

Yours sincerely,

Gábor Kardos

corresponding author

Reviewer 2 Report

Several confounding factors were demonstrated and might be associated with mortality rate. Univariate and multivariate analysis should be performed. Conclusion should relevant to the objectives of the study.

1.There was "unacceptable" and "over" conclusion in the abstract that may mislead the outcome presentation. Line 39-40, the author mentioned CPG adherence was associated with favorable outcomes and 30-day mortality while these results were demonstrated statistical significant. 

2.Several confounding factors were demonstrated and might be associated with mortality rate. Univariate and multivariate analysis should be performed.

3.Conclusion should relevant to the objectives of the study. 

Author Response

Response to Reviewer 2.

We thank the Reviewer for the helpful comments to which we reply as follows.

The changes are highlighted in the manuscript with track changes.

Comment 1: There was "unacceptable" and "over" conclusion in the abstract that may mislead the outcome presentation. Line 39-40, the author mentioned CPG adherence was associated with favorable outcomes and 30-day mortality while these results were demonstrated statistical significant.

Response: Corrections were made as suggested.

Comment 2: Several confounding factors were demonstrated and might be associated with mortality rate. Univariate and multivariate analysis should be performed.

Response:  The results of the univariate analysis are shown in Table 5, while results of the multivariate analysis are shown in Table 6. Yes, there are a lot of confounders. Out of the three factors (increased age, higher CCI score, and higher CRP level) that were associated with higher mortality in the univariate analysis, only the admission CRP level was found to increase the risk of mortality in the multivariate analysis.

Comment 3: Conclusion should relevant to the objectives of the study.

Response: As finally we only analysed mortality rate as patient outcome (and not LOS due to the several confounders), we rephrased study aims to make this clear. We put characteristics of AB use as secondary aim, so no conclusion is necessary for that at the end of the manuscript. 

We hope that these changes have improved the manuscript.

Yours sincerely,

Gábor Kardos

corresponding author

Reviewer 3 Report

Good work. No revisions are necessary.

Author Response

Answer to Reviewer 3.

We thank the Reviewer the time and effort invested in the revision of our manuscript.

Yours sincerely,

Gábor Kardos

corresponding author

Reviewer 4 Report

The authors of this study aimed to evaluate the characteristics and outcome of antibiotic usage in patients admitted to the hospital due to CAP.

According to the authors, the novelty and potential interest for Antibiotics readers is that this is the first field study in Hungary regarding the evaluation of antibiotic prescription patterns for CAP.

Comparisons to studies in other countries, and to national and international guidelines support the main conclusions of this work, which is well organized and highlights the need for stricter (and true adherence) antibiotic stewardship guidelines.

The statistical analysis provided also sustains reported conclusions.

I have made a few comments, aiming to improve the manuscript quality and readability.

Major comments:

  1. The authors should ensure they cite the most recent statistical data in each situation.
  2. Despite considering the conclusion clear and direct, after such a lengthier discussion, I believe the authors should develop, there or in another suitable section, why studies as the one here reported are important and what could be the impact of such work.

Minor comments:

  1. The manuscript is full of sentences or expressions in italics or bold that I do not understand. Some examples: l. 59-60; l. 117-118; l. 188-189; l. 196-199; l. 270-271; among others.
  2. To improve readability, once the authors define an abbreviation (the first time it is mentioned), they should use it continuously and not keep switching between the full word and abbreviation (for instance for CCI; IV; LOS; CRP).
  3. Tables 1 and 3 formatting should be made clearer, as the other tables. Currently, it is confusing to distinguish sub-headers from list items. In Table 2, I advise the authors to replace the highlighted A for adherence for a symbol.
  4. In line 159 where it says “CAP was f common”, I believe the authors only meant “CAP was common”.
  5. The authors should refrain from writing beta-lactam in some situations, and β-lactam in others.
  6. Reference formatting in L. 328 should be corrected.

Author Response

Response to Reviewer 4.

We thank the Reviewer for the helpful comments to which we reply as follows.

The changes are highlighted in the manuscript with track changes.

Major comments:

Comment 1: The authors should ensure they cite the most recent statistical data in each situation.

Response: You are right that some of the reported data could be updated. However, as the study was conducted in 2017, we tried to provide background data close to this date. Hope that this is acceptable.

Comment 2: Despite considering the conclusion clear and direct, after such a lengthier discussion, I believe the authors should develop, there or in another suitable section, why studies as the one here reported are important and what could be the impact of such work.

Response: We added this information as suggested in the end of the discussions.

Minor comments:

Comment 1: The manuscript is full of sentences or expressions in italics or bold that I do not understand. Some examples: l. 59-60; l. 117-118; l. 188-189; l. 196-199; l. 270-271; among others.

Response: We corrected these mistakes as suggested.

Comment 2: To improve readability, once the authors define an abbreviation (the first time it is mentioned), they should use it continuously and not keep switching between the full word and abbreviation (for instance for CCI; IV; LOS; CRP).

Response: We corrected these mistakes as suggested.

Comment 3: Tables 1 and 3 formatting should be made clearer, as the other tables. Currently, it is confusing to distinguish sub-headers from list items. In Table 2, I advise the authors to replace the highlighted A for adherence for a symbol.

Response: We have changed A to symbol “✓” as suggested.

Comment 4: In line 159 where it says “CAP was f common”, I believe the authors only meant “CAP was common”.

Response: We corrected this mistake as suggested.

Comment 5: The authors should refrain from writing beta-lactam in some situations, and β-lactam in others.

Response: We have changed β-lactam to beta-lactam as suggested.

Comment 6: Reference formatting in L. 328 should be corrected.

Response: We added the correct form as suggested.

We hope that these changes have improved the manuscript.

Yours sincerely,

Gábor Kardos

corresponding author

Round 2

Reviewer 2 Report

Appreciated the response to  reviewer from the author.